# Entropy-Reservoir Bregman Projection: An Information-Geometric Unification of Model Collapse

## Abstract

Self-referential learning—training a model on data it generated itself—promises boundless scalability but chronically suffers from *model collapse*: language models degenerate into repetitive text, GANs drop modes, and reinforcement-learning policies over-exploit. Although practitioners employ ad hoc fixes such as real-data mixing, entropy bonuses, knowledge distillation, or retrieval-augmented generation, a single principle that explains both the failure mode and the success of these fixes has remained elusive. We present **Entropy-Reservoir Bregman Projection** (ERBP), an information-geometric framework that unifies these phenomena. We model the closed loop as a stochastic Bregman projection sequence in distribution space. Without external coupling, finite-sample noise forces the system to project onto an ever-shrinking empirical support, causing exponential entropy decay and eventual collapse. Introducing an *Entropy Reservoir*—a high-entropy distribution mixed into each projection—injects a controllable entropy flux that provably stabilises the dynamics. Our theory yields (i) a necessary condition for collapse, (ii) a sufficient condition that guarantees a non-trivial entropy floor, and (iii) closed-form rates that depend only on sample size and the strong-convexity/Lipschitz constants of the Bregman generator. Experiments on large-language-model self-training, Soft Actor-Critic in reinforcement learning, and GAN optimisation validate our predictions and show that disparate stabilisation heuristics correspond to specific reservoir choices and coupling coefficients. ERBP thus transforms a collection of folk remedies into a single, quantitative design rule: monitor and budget your entropy flux.

## 1 Introduction

Modern generative AI, from large language models to diffusion models, is built on the foundation of massive datasets. The paradigm of self-referential learning, where a model iteratively trains on its own generated data, offers a tantalizing solution to the ever-growing demand for data, promising continuous, self-driven improvement (Goodfellow et al., 2014; Schrittwieser et al., 2020).

However, this self-referential loop harbors a fundamental instability. This is not a purely academic concern; it manifests in cutting-edge applications. Consider the "Generative Agents" simulation from Stanford, where 25 AI agents inhabit a virtual town (Park et al., 2023). Initially endowed with rich, human-written backstories, they interact and form memories. Our framework predicts that, being a closed information loop, such a system is destined for **entropy decay**: their language should become formulaic, their behaviors stereotyped, and their unique personalities should fade into shallow caricatures. We define **model collapse** as the overarching degenerative process in recursive learning systems. This general phenomenon manifests in specific domains under different names: as "generative degeneracy" or the "curse of recursion" in LLMs (Holtzman et al., 2019; Shumailov et al., 2023); as the classic problem of **mode collapse** in GANs (where the generator specifically ignores parts of the data distribution) (Arjovsky et al., 2017); and as "policy collapse" in Reinforcement Learning (Haarnoja et al., 2018).

Intriguingly, a set of seemingly unrelated heuristic techniques has proven effective at mitigating these issues: mixing in real data during LLM fine-tuning, knowledge distillation from a teacher

model, entropy regularization in RL, and even label smoothing in standard supervised learning (Szegedy et al., 2016). The degenerative outcome of purely self-referential training is, to be frank, a widely recognized, almost folkloric observation with deep roots in fields like semi-supervised learning (Yarowsky, 1995; Lee et al., 2013) and even cognitive science's models of language evolution (Kirby, 2001). Yet, this empirical wisdom has remained a collection of disparate cautionary tales. A formal, predictive framework that explains **why** collapse is a near-universal constant–and crucially, **why** these different antidotes are all effective–has been elusive.

This paper addresses this gap by proposing that these dynamics are governed by a single, unifying mathematical principle. Our thesis is that the entire process can be modeled as a sequence of Bregman projections in a probability space. The system's fate–stability or collapse–is determined by its coupling to a high-entropy **Entropy Reservoir** ($P_{\text{res},t}$). Model collapse is the inevitable outcome when the system is decoupled from this reservoir ($\lambda_t \to 0$), trapped in the echo chamber of its own increasingly sparse outputs. Conversely, we argue that all successful stabilization techniques are, in essence, different instantiations of coupling the state distribution to such a reservoir, ensuring a vital influx of diversity.

Our main contributions are:

- We introduce and formalize the **Entropy-Reservoir Bregman Projection framework**, providing a unified language to analyze the dynamics of self-referential learning systems.

- We introduce the concept of the **Entropy Reservoir**, showing that techniques like real data mixing, tool use, and human-in-the-loop feedback are all instantiations of this single mathematical object.

- We provide **rigorous proofs** establishing the necessary conditions for model collapse (in the absence of a reservoir) and sufficient conditions for stability (in its presence).

- We present **empirical validation** across LLM self-training, RL policy iteration, and GAN training, demonstrating the broad applicability and predictive power of our framework.

## 2 RELATED WORK

**Empirical and Theoretical Studies of Model Collapse.** The phenomenon of model collapse has been documented extensively across different domains. In LLMs, early work identified issues of text degeneration like repetition and blandness (Holtzman et al., 2019), with recent studies formalizing how recursive self-training leads to a rapid decline in diversity and quality (Shumailov et al., 2023). In the GAN literature, mode collapse is a foundational challenge, addressed by a vast body of work on alternative divergences like Wasserstein distance (Arjovsky et al., 2017) and stabilization techniques such as unrolled optimization (Metz et al., 2017) and various forms of gradient penalties (Salimans et al., 2016; Kodali et al., 2018). In RL, policy degradation due to insufficient exploration is a classic problem, addressed by techniques that explicitly encourage stochasticity and entropy, dating back to early work in maximum entropy RL (Ziebart et al., 2008) and widely used in modern algorithms like A3C (Mnih et al., 2016) and SAC (Haarnoja et al., 2018). Our work provides a unifying geometric explanation for **why** collapse occurs across these domains and formalizes the solution via the Entropy Reservoir.

**The Historical Roots of Self-Referential Learning.** The core dynamic we model is not new. In semi-supervised learning, the method of self-training or pseudo-labeling (Lee et al., 2013), which has roots in early computational linguistics (Yarowsky, 1995), follows a similar loop: a model makes predictions on unlabeled data, and these predictions are used as new training targets. This process is known to be effective but can also amplify its own mistakes, a direct analogue to model collapse. Furthermore, the field of cognitive science, particularly in language evolution, uses the concept of "iterated learning" to model how language is transmitted through generations of learners (Kirby, 2001; Griffiths & Kalish, 2007). These studies show that such transmission can lead to the spontaneous emergence of linguistic structure but also to a loss of complexity, mirroring the entropy decay we describe. Our framework provides a formal, information-geometric model for these long-observed dynamics.

**Information Geometry and Stabilization Techniques.** Our work is built on the tools of information geometry (Amari & Nagaoka, 2000), where Bregman projections are the cornerstone of algorithms like Mirror Descent and Natural Gradient Descent (Amari, 1998; Beck & Teboulle, 2003). However, we repurpose these tools from one-shot optimization to model a closed-loop dynamical system. A variety of methods are known to stabilize self-referential systems, which we unify as instantiations of an Entropy Reservoir. These include knowledge distillation from a teacher model (Buciluǎ et al., 2006; Hinton et al., 2015) and, more recently, coupling models to external sources of information. This is exemplified by Retrieval-Augmented Generation (RAG) which grounds generation in a textual database (Lewis et al., 2020), tool-using agents that call external APIs (Schick et al., 2023), and human-in-the-loop systems like RLHF that align models with human preferences (Christiano et al., 2017; Ouyang et al., 2022). Our framework reveals that these are not isolated tricks but are unified by the common mechanism of providing an effective Entropy Reservoir.

## 3 THE ENTROPY-RESERVOIR BREGMAN PROJECTION FRAMEWORK

We begin by defining the components of our framework. Let $\mathcal{P}$ be the space of probability distributions. We define the **model manifold** $\mathcal{M} \subset \mathcal{P}$ as the set of all distributions realizable by the system. This manifold can be formed by varying parameters (e.g., $\{P_\theta\}$) or by changing contextual inputs like prompts or memories with fixed parameters (e.g., $\{P_\theta(\cdot|\cdot, M)\}$). The distance or divergence between distributions is measured by a Bregman divergence $B_F(P, Q)$. While the Kullback-Leibler (KL) divergence is a prominent example, our framework and its core theoretical results hold for a broad class of Bregman divergences, as detailed in Appendix D.

**State versus Parameters** Throughout this paper, $P_t$ denotes the effective distribution realized by the system at round $t$. It may be obtained by changing parameters $\theta_t$, by adapting prompts or memories $M_t$ with fixed $\theta$, or by any combination thereof. Consequently, the state $P_t$ lives in the probability space $\mathcal{P}$, not necessarily in a parameter space.

Let $\hat{P}_t$ denote the empirical distribution formed by drawing $m$ samples from the current state $P_t$.

**Definition 1** (Entropy Reservoir). *A sequence of distributions $\{P_{\mathrm{res},t}\}_{t\geq 0}$ is a valid Entropy Reservoir if for all $t$, it satisfies:*

1. ***Support Coverage:*** $\mathrm{supp}(\hat{P}_t) \subseteq \mathrm{supp}(P_{\mathrm{res},t})$.

2. ***Entropy Lower Bound:*** $\mathcal{S}_F(P_{\mathrm{res},t}) \geq s_{\min} > 0$.

The dynamics unfold in a three-step iterative process:

1. **Empirical Sampling (The Echo):** From the current state distribution[1] $P_t$, sample $m$ data points to construct a sparse empirical distribution $\hat{P}_t$.

2. **Mixing with the Reservoir:** Form a regularized target distribution $\bar{Y}_t = (1 - \lambda_t)\hat{P}_t + \lambda_t P_{\mathrm{res},t}$, where $\lambda_t \in [0, 1]$ is the coupling coefficient.

3. **Projection Update:** Update the state distribution by projecting onto the mixed target:
$$P_{t+1} = \arg\min_{P \in \mathcal{M}} B_F(P, \bar{Y}_t).$$

**Time-varying versus Constant Coupling.** Throughout the paper, $\lambda_t$ denotes the *time-varying* coupling coefficient. This formulation accommodates arbitrary scheduling strategies, such as annealing, noisy adaptation, or asymptotic decay (e.g., $\lambda_t = 0.1 + 1/t$). In such cases, a strict minimum may never be attained. Therefore, our theoretical stability results (Section 4.3) rely on the *infimum* as a uniform lower bound:
$$\lambda_{\min} := \inf_{t \geq 0} \lambda_t \quad \text{with} \quad 0 \leq \lambda_{\min} \leq 1.$$

This quantity $\lambda_{\min}$ always exists and suffices to guarantee a non-trivial entropy floor. While the theory supports general $\lambda_t$, our experiments (Section 6) adopt the simplified setting of a **constant coupling** ($\lambda_t \equiv \lambda$), implying $\lambda_{\min} = \lambda$.

---

[1]Throughout the paper, we may abusively call $P_t$ "the model", although strictly speaking it is the behavioral distribution induced by parameters and/or external context like prompts and memories.

**Example: Implicit Projections in AI Agents** This framework directly applies to modern AI agents where the underlying model parameters $\theta$ are frozen. Consider the Stanford Generative Agents (Park et al., 2023), where agents' memories $M_t$ are updated based on their experiences (samples $\hat{P}_t$). This memory update $M_{t+1} = \text{Update}(M_t, \hat{P}_t)$ is an operation in prompt-space that **induces** a transition in probability-space from the old state distribution $P_t(\cdot|\cdot) = P_\theta(\cdot|\cdot, M_t)$ to a new one, $P_{t+1}$. The update's functional goal is to align future behavior with recent experience, thus serving as an **implicit projection** towards the empirical distribution $\hat{P}_t$.

At first glance, this update rule resembles a standard step in optimization algorithms like Mirror Descent. However, this formal similarity belies a fundamental conceptual shift: from using projection as a one-shot optimization tool to employing it as a model for a closed-loop dynamical system. To fully illuminate this distinction, which is central to our thesis, we provide a detailed side-by-side comparison in Appendix B (Table 3).

**Optimization vs. Dynamical–System View.** The full side-by-side comparison table has been moved to Appendix B (Table 3). Here we highlight only the key difference: standard Bregman projection is a one-shot optimisation step, whereas ERBP models a self-referential *closed loop*.

As highlighted in Appendix B, Table 3, the crucial difference lies in the self-referential, closed-loop dynamic. The system's next state depends on its own previous output. This feedback loop is precisely what creates the risk of "echo chamber" effects leading to collapse. The Entropy Reservoir and the coupling coefficient are the essential mechanisms that regulate this feedback loop, ensuring a continuous influx of diversity to prevent the system from spiraling into a degenerate state.

Table 1 shows how several common techniques map to this definition. For a more extensive discussion of the design space for various reservoir types, including their respective advantages and disadvantages, see Appendix C.

Table 1: Common stabilization techniques as instantiations of the Entropy Reservoir.

| Reservoir Type $P_{\text{res},t}$ | Corresponding Strategy |
| --- | --- |
| Uniform Distribution $\mathcal{U}$ | Entropy Regularization / Label Smoothing |
| Real Data Distribution $P_{\text{data}}$ | Mixing with Real Data |
| Human Goal/Knowledge Dist. $P_{\text{human}}$ | Human-in-the-Loop (HITL) / RLHF |
| Teacher Model $P_{\text{teacher}}$ | Knowledge Distillation |
| External Tools (Web Search, APIs) | Tool-Using AI Agents |

## 4 THEORY: ENTROPY DYNAMICS UNDER STOCHASTIC BREGMAN PROJECTION

### 4.1 PRELIMINARIES AND NOTATION

Let $\Delta^n = \{p \in \mathbb{R}^n_{\geq 0} \mid \sum_i p_i = 1\}$ be the probability simplex and $F : \text{int}(\Delta^n) \to \mathbb{R}$ a **Legendre-type** convex potential. Its Bregman divergence and $F$-entropy are defined respectively as:

$$B_F(p, q) := F(p) - F(q) - \langle \nabla F(q), p - q \rangle, \tag{1}$$
$$\mathcal{S}_F(p) := -\langle \nabla F(p), p \rangle. \tag{2}$$

**Assumption 1 (Geometry of $F$).** We assume $F$ is $\sigma_F$-strongly convex with respect to the norm $\|\cdot\|$, satisfying $F(p) \geq F(q) + \langle \nabla F(q), p - q \rangle + \frac{\sigma_F}{2}\|p - q\|^2$. Additionally, $\nabla F$ is $L_F$-Lipschitz continuous, such that $\|\nabla F(p) - \nabla F(q)\|_* \leq L_F\|p - q\|$.

**Self-referential loop.** At round $t$, the system's state distribution is $P_t \in \mathcal{M}$. We draw $m$ i.i.d. samples $\{x_i\}_{i=1}^m \sim P_t$ to form the empirical distribution $\hat{P}_t$ supported on these samples. Given a reservoir $P_{\text{res},t}$ and coupling coefficient $\lambda_t \in [0, 1]$, we form the mixed target:

$$\bar{Y}_t := (1 - \lambda_t)\hat{P}_t + \lambda_t P_{\text{res},t}.$$

**Assumption 2 (Approximate projection on a possibly *non-convex* model manifold).** Below, $\varepsilon_t$ measures the optimisation error of each projection; we set the uniform bound $\varepsilon_{\max} := \sup_t \varepsilon_t$. For each $t$ the learning algorithm outputs a new state distribution $P_{t+1} \in \mathcal{M}$ such that

$$B_F(P_{t+1}, \bar{Y}_t) \ \leq \ \varepsilon_t, \qquad \text{with } 0 \leq \varepsilon_t \leq \varepsilon_{\max}. \tag{3}$$

No convexity of $\mathcal{M}$ is required; $\varepsilon_{\max}$ quantifies optimisation error and covers local minima, early stopping, etc. For convenience denote

$$\kappa \ := \ \sqrt{2\,\varepsilon_{\max}}. \tag{4}$$

The learning algorithm itself may update parameters, prompts, memories, or any mechanism that realises the new distribution $P_{t+1}$. We provide a detailed verification that modern algorithms, including LLM fine-tuning (MLE) and Soft Actor-Critic, satisfy this projection assumption in Appendix G.

Set $C_F(m) := \max\limits_{\substack{p \in \Delta^n \\ |\operatorname{supp}(p)| \leq m}} \mathcal{S}_F(p)$

(Shannon case: $C_F(m) = \log m$) and define $\alpha \ := \ \dfrac{\sigma_F}{\sigma_F + mL_F} \ \in (0, 1]$.

### 4.2 Collapse under Vanishing Reservoir Coupling

**Theorem 1** (Entropy Contraction and Support Degeneracy). *Fix a finite sample size $m$ and let $\lambda_t \equiv 0$. Under Assumptions 1–2,*

$$\mathbb{E}\big[\mathcal{S}_F(P_{t+1}) \mid P_t\big] \ \leq \ (1-\alpha)\,\mathcal{S}_F(P_t) \ + \ \alpha\,C_F(m) \ + \ L_F\,\kappa. \tag{5}$$

*Consequently, as $t \to \infty$, the expected entropy is asymptotically bounded:* $\limsup\limits_{t \to \infty} \mathbb{E}[\mathcal{S}_F(P_t)] \ \leq$ $C_F(m) + \frac{L_F\kappa}{\alpha}$. *This implies that the system's entropy inevitably contracts towards a low-entropy state. Even in the ideal case where $\kappa = 0$, the diversity of this state is fundamentally bounded by the sample size $m$, as indicated by the term $C_F(m)$. This leads to a significant loss of generative richness, a phenomenon we characterize as* functional degeneracy, *rather than a complete collapse to a single mode.*

**Proposition 1** (Quantitative decay rate). *In the same setting as Theorem 1, entropy contracts geometrically:*

$$\left| \mathbb{E}[\mathcal{S}_F(P_{t+1})] - C_F(m) - \tfrac{L_F\kappa}{\alpha} \right| \ \leq \ (1-\alpha)\left| \mathbb{E}[\mathcal{S}_F(P_t)] - C_F(m) - \tfrac{L_F\kappa}{\alpha} \right|.$$

### 4.3 Stability with Positive Reservoir Coupling

**Theorem 2** (Entropy floor via reservoir coupling). *Assume the coupling sequence satisfies $\inf_t \lambda_t = \lambda_{\min} \in (0, 1]$ and that every reservoir instance satisfies $\mathcal{S}_F(P_{\mathrm{res},t}) \geq s_{\min} > 0$. Under Assumptions 1–2, for all $t \geq 0$:*

$$\boxed{\mathcal{S}_F(P_{t+1}) \ \geq \ \lambda_t\,s_{\min} \ - \ L_F\,\kappa \ \geq \ \lambda_{\min}\,s_{\min} \ - \ L_F\,\kappa} \tag{6}$$

*In particular, if $\lambda_{\min}\,s_{\min} > L_F\kappa$, the chain can* never *collapse irrespective of sample size $m$.*

**Proposition 2** (Guaranteed entropy floor). *Inequality equation 6 holds for* every *iterate provided the projection error bound equation 3 is satisfied.*

### 4.4 Discussion and Specialisation to KL

For the negative Shannon entropy potential $F(p) = \sum_i p_i \log p_i$ we have $\sigma_F = L_F = 1$. Then $\alpha = \frac{1}{1+m}$ and Theorems 1–2 reduce to the intuitive statements "entropy drops to $\log m$ without reservoirs, but is pinned above $\lambda_t H(P_{\mathrm{res}}) - \kappa$ (here $H$ denotes ordinary Shannon entropy) with reservoirs". All empirical stabilisation heuristics—label smoothing, data mixing, RL entropy bonuses, RAG, RLHF—merely instantiate the parameter pair $(\lambda_t,\ s_{\min})$.

## 5 UNIFYING VIEW OF EXISTING ALGORITHMS

Our framework provides a lens through which disparate algorithms can be seen as variations of the same underlying process. Table 2 provides a summary, now including modern AI agent architectures.

Table 2: Mapping common algorithms to the ERBP framework.

| Domain | Algorithm | Reservoir $P_{\text{res},t}$ | $\lambda_t$ | Outcome |
|---|---|---|---|---|
| LLM Self-Train | Pure Synthetic Data | None | 0 | Collapse |
| LLM Self-Train | Mix w/ Real Data | $P_{\text{data}}$ | $> 0$ | Stable |
| RL | Greedy Policy Iter. | None | 0 | Policy Collapse |
| RL | SAC / Entropy Reg. | Uniform $\mathcal{U}$ | $> 0$ | Exploration/Stable |
| Supervised | Label Smoothing | Uniform $\mathcal{U}$ | $\eta$ (fixed) | One-shot Regularization |
| Generative Agents | Stanford Town (Memory) | None (post-init) | $\approx 0$ | Behavioral Collapse |
| Interactive Agents | Tool Use / HITL / RLHF | Web, APIs, $P_{\text{human}}$ | $> 0$ | Sustained Problem Solving |

It is particularly insightful to view **Label Smoothing** as a single-step, open-loop special case. The "self-sampling" is replaced by drawing a batch from a fixed dataset, and the one-hot labels are mixed with a uniform distribution (the reservoir). The system executes for only one projection step. Because the loop is not closed, the long-term stability problem of model collapse does not arise. This highlights how the same mechanism serves regularization in an open-loop system and ensures survival in a closed-loop one.

## 6 EXPERIMENTS

We validate the ERBP framework across three modalities: language modeling, image generation, and continuous control. We focus on validating the theoretical predictions of entropy contraction (Thm. 1) and reservoir stability (Thm. 2). Detailed experimental setups, hyperparameters, and extended analyses are provided in Appendix F.

In all experiments, we employ a constant coupling coefficient, denoted simply as $\lambda$ (i.e., $\lambda_t \equiv \lambda$).

### 6.1 LANGUAGE MODELING: ENTROPY DECAY AND COLLAPSE DIMENSIONS

**Validating Theorems 1 & 2 (Exp 1).** We first simulated a closed-loop agent using `distilgpt2`. As shown in Figure 1, the system exhibits the predicted bifurcation. Without a reservoir ($\lambda_t \equiv 0$), unique n-gram counts (a proxy for $\mathcal{S}_F$) decay exponentially, confirming the contraction bound in Theorem 1. Conversely, coupling with a high-entropy reservoir ($\lambda_t > 0$) maintains diversity, empirically verifying the entropy floor guaranteed by Theorem 2.

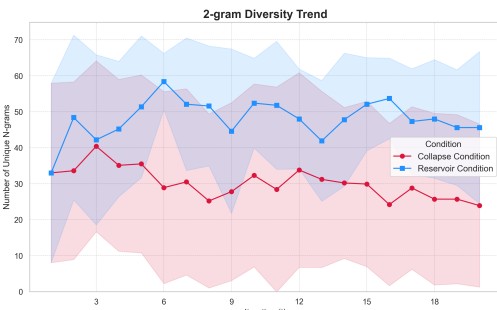

Figure 1: Exp 1 Results. **Collapse** ($\lambda_t \equiv 0$) leads to rapid entropy decay. **Reservoir** ($\lambda_t > 0$) stabilizes diversity, validating Thm. 2.

**Two Dimensions of Collapse (Exp 2).** To dissect the nature of collapse, we simulate a recursive self-training loop using `distilgpt2`, where the model is *continuously fine-tuned* on outputs gen-

erated from a fixed set of prompts (e.g., "The", "In"). We analyzed this process under Greedy vs. Stochastic Sampling ($k = 20$, temperature $\tau = 0.7$) decoding strategies (Figure 2). The results reveal that collapse manifests orthogonally as *Knowledge Collapse* (divergence from ground truth, high PPL) and *Functional Degeneracy* (support contraction, low unique bigram ratio). Notably, sampling strategies without real data ($\lambda_t \equiv 0$) maintain non-zero diversity—validating the $C_F(m)$ term in Thm. 1—but suffer from exploding PPL, indicating a "random walk" on a degraded manifold. In contrast, the reservoir strategy ($\lambda_t \equiv 0.1$), implemented via fixed-budget batch mixing, successfully mitigates both, preserving low PPL and high diversity.

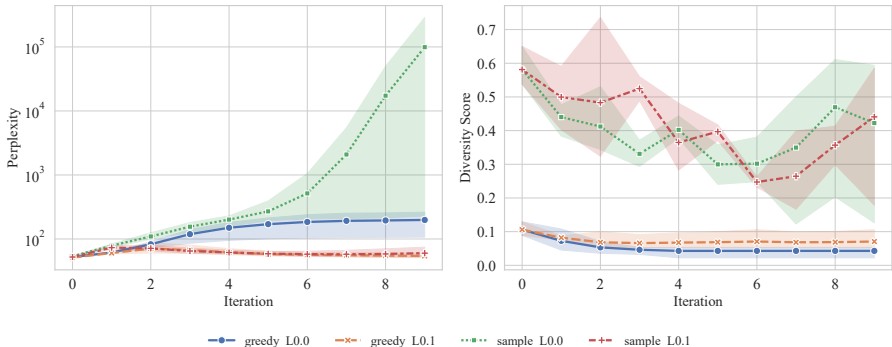

Figure 2: Exp 2: The decoupling of Knowledge Collapse (PPL) and Functional Degeneracy (Diversity). Without a reservoir, models either freeze (Greedy) or hallucinate (Sample), validating Eq. 5.

## 6.2 GENERATIVE IMAGE SYNTHESIS AND CONTROL DYNAMICS

**Recursive GAN Training (Exp 3).** We trained a recursive GAN on MNIST for $T = 60$ generations. Figure 3 highlights a critical failure mode: internal adversarial losses ($\mathcal{L}_G, \mathcal{L}_D$) remain low even during collapse, failing to detect the degradation. However, once we define *Oracle Entropy* as $\mathcal{H}_{\text{oracle}} := -\sum_c \hat{p}(c) \log \hat{p}(c)$, where $\hat{p}(c)$ is the marginal class distribution predicted by a frozen classifier over a generated batch, the Oracle Entropy metric reveals catastrophic mode collapse for $\lambda_t \equiv 0$.

Visual inspection confirms this divergence. As shown in Figure 4, the uncoupled generator ($\lambda = 0$) degenerates into producing indistinguishable blurs at Generation 60. In contrast, the reservoir-coupled generator ($\lambda = 0.2$) acts as a distributional anchor, forcing the system to maintain global diversity. This prevents the discriminator from adapting to degenerate data, preserving clearly recognizable digit structures.

**Geometric Stability in Reinforcement Learning (Exp 4).** To test the hypothesis that entropy reservoirs prevent topological entrapment, we designed a continuous control environment with a *Double Well* reward landscape. The state space is 1D continuous ($x \in [-10, 10]$), containing a deceptive local optimum at $x = -2$ and a global optimum at $x = 4$, separated by a low-reward valley. We trained Soft Actor-Critic (SAC) agents for 10,000 steps.

Figure 5 illustrates the bifurcation in learning dynamics.

- **Collapse Regime ($\lambda \to 0$):** Agents with negligible entropy coupling exhibit rapid entropy decay (Thm. 1). Geometrically, the policy distribution contracts to a point mass immediately. Lacking the "volume" to traverse the probability manifold, the gradient flow becomes trapped, permanently locking the agent into the sub-optimal local maximum at $x = -2$.

- **Reservoir Regime ($\lambda = 0.2$):** Coupling the policy to a uniform entropy reservoir enforces the lower bound from Thm. 2. This maintained variance acts as a geometric regularizer, smoothing the effective optimization landscape. The agent retains sufficient distributional width to bridge the valley, consistently converging to the global optimum at $x = 4$.

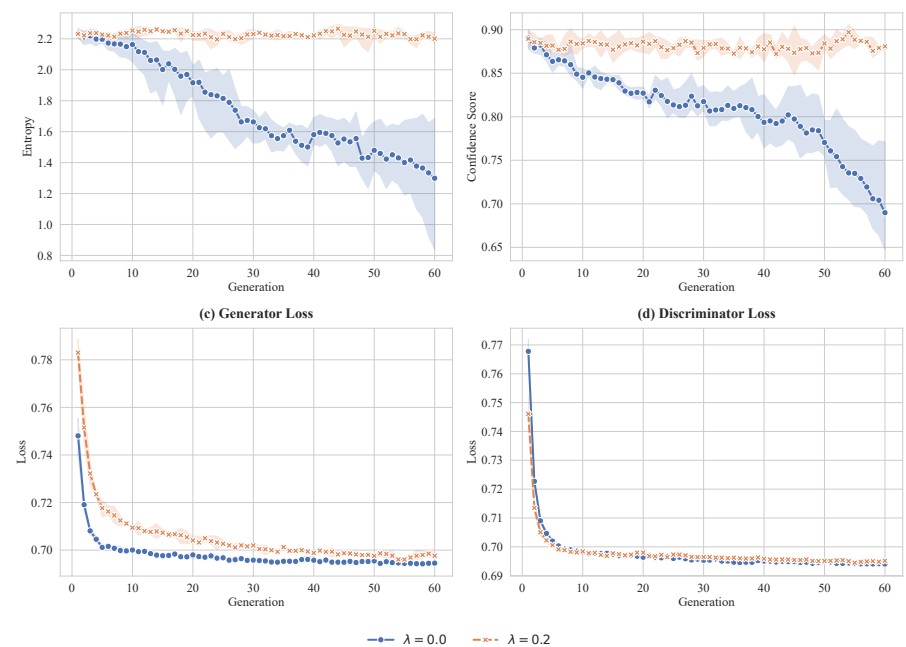

Figure 3: Exp 3: GAN metrics over 60 generations. Internal losses fail to signal collapse; only the external entropy metric (Oracle) reveals the stabilizing necessity of the Reservoir.

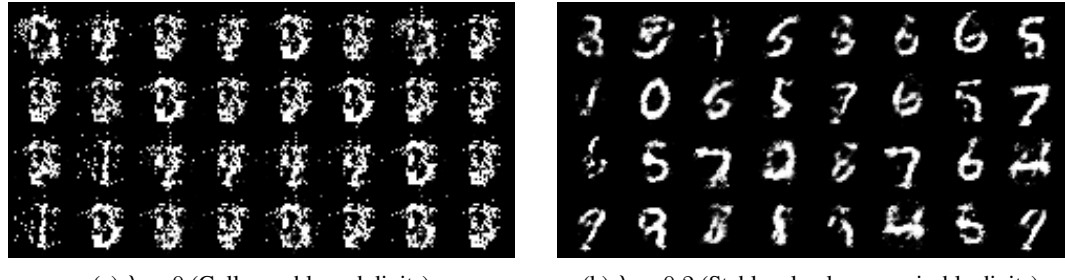

(a) $\lambda = 0$ (Collapse, blurred digits)    (b) $\lambda = 0.2$ (Stable, clearly recognisable digits)

Figure 4: Visual comparison of GAN outputs at Generation 60. Without a reservoir ($\lambda = 0$), the generator suffers catastrophic mode collapse and quality degradation. With a reservoir ($\lambda = 0.2$), digit structure is preserved.

# 7 DISCUSSION AND CONCLUSION

## 7.1 GENERALITY OF PROJECTIONS IN DISTRIBUTION SPACE

As established in Section 3, our framework's core objects are probability distributions, not parameter vectors. This distinction is crucial for understanding its generality. One might have argued that systems like the Stanford Generative Agents (Park et al., 2023) differ from traditional training, as the underlying model parameters $P_\theta$ are frozen and only the agents' memory-prompts $M_t$ are updated. This distinction, in fact, highlights the profound generality of our framework.

Our theory operates in the **space of probability distributions**, not a specific parameter space. The system's state at time $t$ is the effective conditional distribution $P_t(\cdot|\cdot) := P_\theta(\cdot|\cdot, M_t)$. The set of all distributions reachable by varying the prompt $M$ forms a specific submanifold, $\mathcal{M}_\theta$, within the space of all possible distributions.

The daily memory update, $M_{t+1} = \text{Update}(M_t, \hat{P}_t)$, is an operation in prompt-space. However, it **induces a transition** in probability-space from $P_t$ to $P_{t+1}$. The functional purpose of this update is to make the agent's future behavior ($P_{t+1}$) reflect its recent experiences ($\hat{P}_t$). Therefore, this update serves as an **implicit projection step**. It moves the system's distribution along the manifold $\mathcal{M}_\theta$

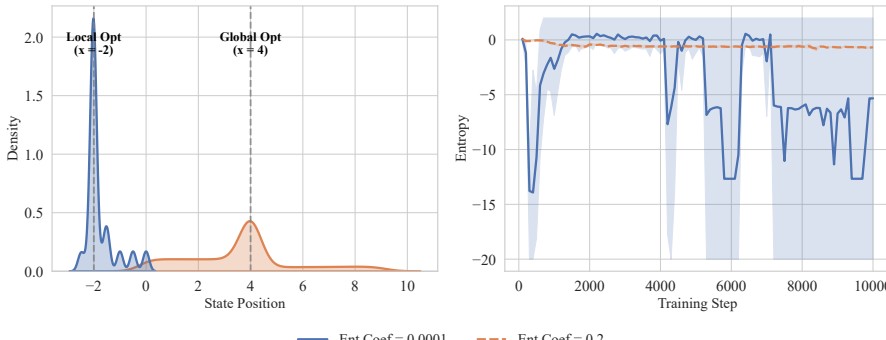

Figure 5: Exp 4: **(Left)** Final state distributions. Without a reservoir ($\lambda \approx 0$), policies collapse into the local optimum at $x = -2$. The reservoir ($\lambda = 0.2$) enables the agent to traverse the valley and lock onto the global optimum at $x = 4$. **(Right)** Entropy evolution showing the predicted floor effect.

towards the empirical distribution of its own interactions. The core dynamics of entropy decay thus remain unchanged, as the system is still projecting onto its own sparse samples without an external entropy source.

## 7.2 THE PARADOX OF REFLECTION: TOOL USE AND HUMAN FEEDBACK AS NECESSARY RESERVOIRS

Our framework offers a stark explanation for why purely introspective AI agents, which "reflect" on their own outputs to refine their plans, often fall into "cognitive loops" or reasoning dead-ends. This process can be modeled as a sequence of self-projections with $\lambda_t = 0$. An agent's "reflection" is a sample $\hat{P}_t$ from its current belief state $P_t$. Incorporating this reflection is a projection that updates $P_t$ to $P_{t+1}$. If $P_t$ contains a flawed belief, the reflection will likely reinforce it, leading to an exponential decay of cognitive diversity and a collapse into a set of dogmatic, incorrect beliefs.

The antidote, as predicted by our theory, is the introduction of an Entropy Reservoir. In the context of AI agents, this coupling is achieved through two primary mechanisms: **automated tool use** and **human-in-the-loop feedback**.

- **Automated Tools** (e.g., web search, APIs) act as information-geometric operations that couple the agent's internal state with high-entropy, high-fidelity external data distributions ($P_{\text{data}}$). This is the principle behind Retrieval-Augmented Generation (Lewis et al., 2020) and tool-forming models (Schick et al., 2023).

- **Human Feedback**, whether through explicit correction, preference tuning (as in RLHF (Ouyang et al., 2022)), or direct instruction, represents a coupling with the highest-fidelity entropy reservoir available–the user's own goal distribution, $P_{\text{human}}$. The foundations for this were laid by learning from human preferences (Christiano et al., 2017).

This provides the necessary influx of new information to break internal feedback loops, correct flawed reasoning, and ensure alignment. This leads to a fundamental design principle: **an agent's effective intelligence is limited not by its capacity for self-reflection, but by the bandwidth and quality of the entropy reservoirs–both automated and human–it is coupled with.**

## 7.3 A GEOMETRIC RE-INTERPRETATION OF CLASSICAL REGULARIZERS

Label Smoothing (LS) is traditionally justified as a way to prevent over-confidence. Within our framework it becomes the *one-shot* analogue of reservoir coupling. Analytical work has shown LS encourages better feature representations (Müller et al., 2019), which aligns with our geometric view:

- The data label $\mathbf{e}_y$ (a simplex vertex) is replaced by $\bar{y} = (1 - \eta)\mathbf{e}_y + \eta\mathcal{U}$.
- The projection $P^\star = \arg\min_{P \in \mathcal{M}} B_F(P, \bar{y})$ inevitably lands in the *interior* of the simplex, injecting an entropy budget $\eta \log |\mathcal{Y}|$.

Thus the same geometric mechanism that keeps a closed loop from collapsing is what makes LS improve generalisation in an open loop.

### 7.4 Limitations and Future Work

**On the Tightness of the Collapse Bound.** Our analysis provides a general bound on the rate of entropy decay (Theorem 1). However, in practice, collapse can occur much faster than a large sample size $m$ would suggest. This is not a contradiction, but highlights that our bound could be tightened by accounting for the specifics of the sampling process. Decoding strategies such as top-k, nucleus, or low-temperature sampling create an effective sampling distribution that is already much "sharper" than the original model distribution $P_t$. A promising direction is to derive more precise bounds by explicitly modeling these decoding strategies.

**Characterizing the Steady State.** Our stability result (Theorem 2) provides a crucial "survival guarantee" but does not fully characterize the long-term asymptotic behavior. A natural next question is whether the system converges to a stable fixed point, a limit cycle, or exhibits more complex dynamics. Analyzing the existence and uniqueness of solutions to the self-referential fixed-point equation, $P^* = \arg\min_{P \in \mathcal{M}} B_F(P, (1 - \lambda_t)P^* + \lambda_t P_{\text{res}})$, is a significant theoretical undertaking and a key direction for future research.

**Projection error $\varepsilon_{\max}$.** We assumed either exact projection or a uniform error bound $\varepsilon_{\max}$. Characterising how optimiser noise and other sources of projection error compound over many rounds is an important practical question.

**Non-convex model manifolds.** Deep networks induce highly non-convex model manifolds $\mathcal{M}$. While Bregman projection is still well-defined via empirical risk minimisation, global convergence is not guaranteed. As empirically observed in our RL experiments (Sec. 6.2), the reservoir helps smooth the optimization trajectory, preventing collapse into singular local attractors. Formalising how SGD noise interacts with this reservoir effect to ensure global convergence remains an important theoretical challenge.

**Adaptive Coupling.** A fixed coupling coefficient $\lambda_t$ is often sub-optimal. Future work could explore adaptive coupling strategies, such as an entropy-feedback annealing scheme that dynamically adjusts $\lambda_t$ based on the system's current entropy deficit, allowing for more fine-grained control over the stability-fidelity trade-off.

**Connection to Continual Learning.** Our framework offers a new lens for continual learning, where catastrophic forgetting bears a strong resemblance to model collapse. Future work could formalize this connection, viewing techniques like Elastic Weight Consolidation (EWC) as implicit entropy reservoirs designed to preserve the knowledge (entropy) of past tasks.

### 7.5 Conclusion

We revisited the long-standing folk wisdom—"self-training collapses without real data"—and placed it on firm information-geometric ground. By framing self-referential learning as a chain of stochastic Bregman projections, we showed that entropy influx from an external reservoir is both *necessary* and (almost) *sufficient* for long-term stability.

Our empirical results across LLMs, GANs, and Reinforcement Learning demonstrate that this is not merely a modality-specific issue, but a universal law of closed-loop information processing. The quantitative bounds and design guidelines provided here transform this insight into a practical tool: an *entropy budget* that can be monitored and actively controlled. We hope this perspective will catalyse new algorithms that treat entropy not as an afterthought, but as a first-class resource in modern machine learning systems.

### Acknowledgments

The authors acknowledge the use of Gemini and ChatGPT to assist with language editing and polishing of the text. The conceptualization, theoretical derivations, and experimental design are entirely the original work of the authors.

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

# A  PROOFS FOR SECTION 4

## A.1  FORMAL DEFINITIONS

For completeness, we provide the formal definitions of the geometric properties required in Assumption 1.

**Definition 2** ($\sigma_F$-Strong Convexity). *A differentiable function $F : \Omega \to \mathbb{R}$ is $\sigma_F$-strongly convex with respect to a norm $\| \cdot \|$ if for all $x, y \in \Omega$:*

$$F(y) \geq F(x) + \langle \nabla F(x), y - x \rangle + \frac{\sigma_F}{2} \|y - x\|^2. \tag{7}$$

**Definition 3** ($L_F$-Lipschitz Smoothness). *The gradient $\nabla F$ is $L_F$-Lipschitz continuous with respect to a dual norm $\| \cdot \|_*$ if for all $x, y \in \Omega$:*

$$\|\nabla F(x) - \nabla F(y)\|_* \leq L_F \|x - y\|. \tag{8}$$

## A.2  TECHNICAL LEMMAS

**Lemma 1** (Sampling entropy bound). *For the empirical law $\hat{P}_t$ of $m$ i.i.d. draws, $\mathcal{S}_F(\hat{P}_t) \leq C_F(m)$.*

**Lemma 2** (Entropy–divergence continuity). *For all $p, q \in \Delta^n$, $\left| \mathcal{S}_F(p) - \mathcal{S}_F(q) \right| \leq L_F \sqrt{2 \, B_F(p, q)}$.*

*Proof sketch.* $\mathcal{S}_F(p) - \mathcal{S}_F(q) = \langle \nabla F(q) - \nabla F(p), \, p - q \rangle$. Apply Cauchy–Schwarz and Lipschitzness of $\nabla F$, then use strong convexity to turn $\|p - q\|_2$ into $\sqrt{B_F(p, q)}$. $\square$

### A.3 PROOF OF THEOREM 1

**Condition on $P_t$. Step 1.** $\mathbb{E}[\hat{P}_t] = P_t$ and $\mathbb{E}\|\hat{P}_t - P_t\|_2^2 \leq \frac{1}{m}$. Strong convexity gives $\mathbb{E}[B_F(\hat{P}_t, P_t)] \geq \frac{\sigma_F}{2m}$. Lemma 2 yields

$$\mathbb{E}\big[\mathcal{S}_F(P_t) - \mathcal{S}_F(\hat{P}_t) \mid P_t\big] \geq \frac{\sigma_F}{L_F}\,\mathbb{E}[B_F(\hat{P}_t, P_t)] \geq \frac{\sigma_F}{2L_F m}.$$

Using $\mathcal{S}_F(\hat{P}_t) \leq C_F(m)$ (Lemma 1) and rearranging gives

$$\mathbb{E}[\mathcal{S}_F(\hat{P}_t) \mid P_t] \leq (1-\alpha)\mathcal{S}_F(P_t) + \alpha C_F(m),$$

with $\alpha = \sigma_F/(\sigma_F + mL_F)$.

**Step 2 (projection loss).** By equation 3 and Lemma 2, $\big|\mathcal{S}_F(P_{t+1}) - \mathcal{S}_F(\hat{P}_t)\big| \leq L_F\kappa$. Combine with Step 1 to obtain equation 5. Iterating the affine contraction yields the expectation bound; martingale convergence plus the support argument of the main text finishes the proof.

### A.4 PROOF OF PROPOSITION 1

Iterate equation 5 and observe that the constant $C_F(m) + \frac{L_F\kappa}{\alpha}$ is a fixed point of the affine map.

### A.5 PROOF OF THEOREM 2

By the convexity of the potential $F$ (and consequently the concavity of entropy $\mathcal{S}_F$), the entropy of the mixture is lower-bounded by the weighted entropy of its components:

$$\mathcal{S}_F(\bar{Y}_t) = \mathcal{S}_F((1-\lambda_t)\hat{P}_t + \lambda_t P_{\text{res},t}) \geq (1-\lambda_t)\mathcal{S}_F(\hat{P}_t) + \lambda_t\,\mathcal{S}_F(P_{\text{res},t}).$$

Since entropy is non-negative (for discrete Shannon entropy) or generally bounded from below in our setting, and specifically focusing on the reservoir's contribution, we have the looser but sufficient bound:

$$\mathcal{S}_F(\bar{Y}_t) \geq \lambda_t\,\mathcal{S}_F(P_{\text{res},t}) \geq \lambda_t s_{\min}.$$

Combining this with the projection error bound from Eq. equation 3 and Lemma 2:

$$\mathcal{S}_F(P_{t+1}) \geq \mathcal{S}_F(\bar{Y}_t) - L_F\kappa \geq \lambda_t s_{\min} - L_F\kappa.$$

Taking the infimum over $t$ yields the global floor $\lambda_{\min}s_{\min} - L_F\kappa$.

### A.6 PROOF OF PROPOSITION 2

Immediate from the bound in Theorem 2.

### A.7 ADDITIONAL REMARKS ON THE DIVERGENCE FAMILY

The foregoing proofs use only Assumptions 1–2 and therefore extend to any Bregman generator listed in Table 5. For each generator evaluate $(\sigma_F, L_F)$ to plug into the bounds. Notably, for squared-$\ell_2$ loss $\sigma_F = L_F = 1$ just like KL, while for the $\alpha$-divergence both constants scale with the exponent $\alpha$.

## B   DETAILED OPTIMISATION–DYNAMICS COMPARISON

## C   EXTENDED DESIGN SPACE FOR ENTROPY RESERVOIRS

Table 4 extends the catalogue of possible entropy reservoirs already given in the main text, providing a broader design space for practitioners.

Table 3: Comparison of Bregman projection as an optimisation step versus a dynamical-system model.

| Feature | Standard Bregman Projection (optimisation) | ERBP (dynamical system) |
|---|---|---|
| Core objective | Constrained optimisation; distance minimisation. | Long-term dynamical stability analysis. |
| Projection target | Static / exogenous $\hat{P}$. | Dynamic mixture $\bar{Y}_t = (1 - \lambda_t)\hat{P}_t + \lambda_t P_{\text{res}}$. |
| Information flow | Open loop. | Closed loop, self-referential. |
| Key challenge | Computational solvability. | Entropy preservation; avoiding collapse. |
| Long-term behaviour | Iterating without reservoir $\Rightarrow$ entropy dissipation. | Positive $\lambda_t$ guarantees entropy floor. |

Table 4: Extended design space for $P_{\text{res},t}$.

| Reservoir | Pros / Cons and Typical Use |
|---|---|
| Uniform $\mathcal{U}$ | Maximum entropy; analytic; but can introduce label–feature mismatch. |
| Real data $P_{\text{data}}$ | High fidelity; requires external corpus; legal/privacy constraints. |
| Human feedback $P_{\text{human}}$ | Highest fidelity for alignment; gold standard; expensive to acquire. |
| Snapshot ensemble $\{P_{t-k}\}$ | No extra data; cheap; but entropy gain fades as snapshots converge. |
| High-temperature $P_t^{(\tau)}$ | Preserves semantics while flattening modes; $\tau$ is tunable. |
| Retrieval-augmented mixture | Contextual diversity; naturally scales with RAG pipelines. |

## D  GENERALITY OF THE FRAMEWORK ACROSS BREGMAN DIVERGENCES

While the main text uses Shannon entropy (associated with KL divergence) to build intuition, our core results on collapse and stability are not restricted to KL divergence. They hold for any Bregman divergence $B_F(p,q) = F(p) - F(q) - \langle \nabla F(q), p - q \rangle$ generated by a Legendre-type convex function $F$. This generality stems from two facts:

1. The proofs of our theorems rely on the generalized Pythagorean theorem for Bregman divergences, a property that holds for any such divergence, not just KL.

2. The concept of entropy can be generalized. For any generator $F$, we can define a generalized entropy function $\mathcal{S}_F(p) = -\langle \nabla F(p), p \rangle$. For the negative Shannon entropy generator $F(p) = \sum p_i \log p_i$, this definition recovers the standard Shannon entropy $\mathcal{S}_F(p) = H(p)$.

The stability condition (Theorem 2) relies on the fact that mixing with a reservoir provides a lower bound on the entropy of the target. This property also generalizes. Due to the convexity of $F$, it can be shown that the generalized entropy of the mixed target is lower-bounded by the reservoir's entropy: $\mathcal{S}_F((1 - \lambda_t)p + \lambda_t r) \geq \lambda_t \mathcal{S}_F(r)$.

This generalization extends directly to our quantitative results. The collapse rate and stability bounds can be expressed in terms of the geometric properties of the potential function $F$. If $F$ is $\sigma_F$-strongly convex and its gradient $\nabla F$ is $L_F$-Lipschitz on the probability simplex, then the quantitative bounds take the following general form:

- **Generalized Decay Rate** ($\lambda_t = 0$)**:** The decay of the generalized entropy is governed by:

$$\mathbb{E}[\mathcal{S}_F(P_{t+1})|P_t] \leq \left(1 - \frac{\sigma_F}{\sigma_F + mL_F}\right)\mathcal{S}_F(P_t)$$

- **Generalized Stability Bound** ($\lambda_t > 0$)**:** The stability bound becomes:

$$\mathcal{S}_F(P_{t+1}) \geq \lambda_t \mathcal{S}_F(P_{\text{res}}) - L_F \sqrt{2B_F(P_{t+1}, \bar{Y}_t)}$$

These inequalities demonstrate that the core dynamics of entropy decay and stabilization are not artifacts of KL divergence but are fundamental consequences of the Bregman projection geometry.

Table 5: The Bregman divergence family and their associated algorithms.

| Potential Function $F(x)$ | Bregman Divergence $B_F$ | Covered Domains / Algorithms |
|---|---|---|
| $\sum x_i \log x_i$ (Neg. Entropy) | KL Divergence ($D_{KL}(p\|q)$) | MLE, REINFORCE, Self-training LLMs (mode-covering) |
| $-\sum \log x_i$ | Reverse KL ($D_{KL}(q\|p)$) | Early GANs, some RL (mode-seeking) |
| $\frac{1}{2}\|x\|^2$ | Squared Euclidean ($L_2^2$) | Autoencoders, VAEs, Diffusion Models (continuous data) |
| $\frac{1}{\alpha(\alpha-1)} \sum x_i^\alpha$ | $\alpha$-divergence | Power EP, Variational-$\alpha$ (unifies mode-seeking/covering) |
| $\frac{1}{\beta(\beta+1)} \sum (x_i^{\beta+1} - x_i)$ | $\beta$-divergence | Sparse Coding, NMF (bridges KL and IS divergence) |

## E   NOTATION SUMMARY

Table 6: Summary of key notation.

| Symbol | Description |
|---|---|
| $P_t$ | The system's state/behavioral distribution at time $t$. Lives in $\mathcal{P}$. |
| $\theta_t$ | The vector of model parameters at time $t$. |
| $M_t$ | The memory, prompt, or other context at time $t$. |
| $\mathcal{P}$ | The space of all probability distributions. |
| $\mathcal{M}$ | The model manifold; the subset of $\mathcal{P}$ realizable by the system. |
| $\hat{P}_t$ | The empirical distribution from $m$ samples of $P_t$. |
| $P_{\text{res},t}$ | The entropy reservoir distribution at time $t$. |
| $\bar{Y}_t$ | The mixed target distribution for the projection step. |
| $\lambda_t$ | The reservoir coupling coefficient at time $t$. |
| $B_F(P,Q)$ | The Bregman divergence from $Q$ to $P$ generated by potential $F$. |
| $\mathcal{S}_F(P)$ | The generalized $F$-entropy of a distribution $P$. |

## F   DETAILED EXPERIMENTAL SETUP AND ADDITIONAL RESULTS

### F.1   EXPERIMENT 1: FROZEN LLM SIMULATION

**Setup.** We used the `distilgpt2` model. Ten independent trials (20 iterations each) were run.

- **Collapse Condition** ($\lambda_t \equiv 0$)**:** Next prompt = previous model output.

- **Reservoir Condition** ($\lambda_t > 0$)**:** Next prompt = previous output + high-entropy sentence from external text.

Generation parameters: nucleus sampling ($p = 0.95$), top-k ($k = 50$), max new tokens 75.

**Extended Results.** Table 7 quantifies the final state divergence.

Table 7: Avg. unique n-gram counts at $t = 20$ (Mean $\pm$ Std).

| Condition | Bigrams | Trigrams |
|---|---|---|
| Collapse ($\lambda_t \equiv 0$) | $23.9 \pm 22.7$ | $24.6 \pm 23.4$ |
| Reservoir ($\lambda_t > 0$) | $45.6 \pm 21.2$ | $47.0 \pm 21.7$ |

## F.2 EXPERIMENT 2: LLM SELF-TRAINING

**Implementation Setup.** We utilized the Hugging Face implementation of `distilgpt2`. The model undergoes continuous fine-tuning for a total of $T = 50$ iterations. In each iteration, the model is trained for 1 epoch on the mixed dataset using the AdamW optimizer with a learning rate of $5 \times 10^{-5}$. Unlike unconditional generation, we seeded the generation process to simulate sentence completion tasks using a fixed set of 7 distinct prefixes: `["The", "In", "It", "A", "Once", "However", "Despite"]`, with a maximum generation length of 50 tokens. We monitored performance using two metrics: **Perplexity (PPL)** on a held-out Wikitext-2 test set to measure distribution modeling, and **Diversity** (Unique Bigram Ratio) to quantify the richness of generated text and detect local repetition.

**Experimental Design.** We conducted a $2 \times 2$ factorial experiment: $\lambda_t \in \{0, 0.1\} \times$ {Greedy, Sample}, to evaluate the interplay between decoding stochasticity and reservoir mixing:

- **Greedy Decoding:** Deterministic search (temperature $\tau \to 0$), which minimizes local entropy ($m \to 1$).
- **Stochastic Sampling:** top-$k$ sampling ($k = 20$, temperature $\tau = 0.7$), which introduces randomness (Effective $m > 1$).

**Quantitative Analysis.** Table 8 presents the final metrics at Iteration 9. The results confirm that while sampling delays the appearance of repetition (Diversity stays $> 0$ compared to Greedy's near-zero diversity), it does not prevent the loss of ground-truth probability mass without a reservoir. Specifically, the explosion in PPL for the Sampling method with $\lambda = 0$ confirms the "random walk" hypothesis: without the anchor of real data, the model drifts aimlessly away from the true manifold. In contrast, the reservoir ($\lambda = 0.1$) successfully stabilizes both metrics, maintaining low PPL and healthy diversity.

Table 8: Final LLM Performance at Iteration 9 (Mean $\pm$ Std).

| Method | $\lambda$ | Iter | PPL | Diversity |
|---|---|---|---|---|
| Greedy | 0 | 9 | $198.38 \pm 81.29$ | $0.0429 \pm 0.0178$ |
| Greedy | 0.1 | 9 | $\mathbf{54.38 \pm 4.83}$ | $0.0707 \pm 0.0304$ |
| Sample | 0 | 9 | $99108.09 \pm 162891.82$ | $0.4224 \pm 0.2573$ |
| Sample | 0.1 | 9 | $\mathbf{60.04 \pm 12.58}$ | $\mathbf{0.4408 \pm 0.2272}$ |

## F.3 EXPERIMENT 3: RECURSIVE GANS

**Setup.** Dataset: MNIST. Loop: 60 generations. Training data construction: $\mathcal{D}_{train}^{(t+1)} = (1 - \lambda_t) \cdot G_t(z) + \lambda_t \cdot \mathcal{D}_{reservoir}$.

- **Control ($\lambda_t \equiv 0$):** Pure synthetic loop.
- **Exp ($\lambda_t \equiv 0.2$):** 20% real data mixing.

**Oracle Metric.** A pre-trained CNN classifier (fixed weights) was used to measure the entropy of the generated class distribution, providing an objective measure independent of the discriminator $D_t$.

**Quantitative Analysis.** Table 9 shows the collapse in Oracle Entropy for the uncoupled system. While the visual results in the main text (Figure 4) show the qualitative difference, the table below quantifies the severity of the collapse: the confidence of the collapsed model drops significantly, and the entropy of the class distribution is nearly halved.

Table 9: Final GAN Metrics at Generation 60 (Mean $\pm$ Std).

| $\lambda$ | Gen | Oracle Entropy | Confidence |
|---|---|---|---|
| 0 | 60 | $1.2991 \pm 0.4282$ | $0.6898 \pm 0.0704$ |
| 0.2 | 60 | $\mathbf{2.2011 \pm 0.0338}$ | $\mathbf{0.8809 \pm 0.0030}$ |

## F.4 Experiment 4: Geometric Stability in Reinforcement Learning

**Setup: The Double-Well Potential.** To test the hypothesis that entropy reservoirs prevent topological entrapment, we designed a continuous control environment with a *Double Well* reward landscape. The state space is 1D continuous ($x \in [-10, 10]$). The reward function contains a deceptive local optimum at $x = -2$ (reward $\approx 1.0$) and a global optimum at $x = 4$ (reward $\approx 10.0$), separated by a low-reward valley. We trained Soft Actor-Critic (SAC) agents for 10,000 steps, mapping the algorithm's entropy coefficient directly to our framework's coupling parameter $\lambda$.

**Results: Avoiding Local Attractors.** Figure 5 illustrates the bifurcation in learning dynamics.

- **Collapse Regime ($\lambda \to 0$):** Agents with negligible entropy coupling (blue lines) exhibit rapid entropy decay, validating Theorem 1. Geometrically, the policy distribution contracts to a point mass almost immediately. Lacking the "volume" to traverse the probability manifold, the gradient flow becomes trapped in the nearest basin of attraction, permanently locking the agent into the sub-optimal local maximum at $x = -2$.

- **Reservoir Regime ($\lambda = 0.2$):** Coupling the policy to a uniform entropy reservoir (orange lines) enforces the lower bound from Theorem 2. This maintained variance acts as a geometric regularizer, smoothing the effective optimization landscape. The agent retains sufficient distributional width to bridge the low-reward valley, consistently converging to the global optimum at $x = 4$.

This experiment confirms that the Entropy Reservoir is not merely a noise injector, but a topological necessity for non-convex optimization in self-referential loops.

## G Concrete Learning Algorithms that Satisfy Assumption 2

Assumption 2 requires that the learning algorithm performs an approximate projection onto the model manifold $\mathcal{M}$ with respect to the Bregman divergence generated by $F$. Formally, $P_{t+1} \approx \arg\min_{P \in \mathcal{M}} B_F(P, \bar{Y}_t)$. In this section, we demonstrate that three major classes of modern learning algorithms—Large Language Models (LLMs), Mean Squared Error (MSE) regression, and Soft Actor-Critic (SAC)—satisfy this assumption.

### G.1 Large Language Models (Maximum Likelihood Estimation)

Standard LLM training minimizes the cross-entropy loss, which is equivalent to minimizing the *Forward KL divergence* $\mathrm{KL}(\bar{Y}_t \| P)$. However, our Assumption 2 (specifically $B_F(P, \bar{Y}_t)$ with neg-entropy potential) relies on the *Reverse KL divergence* $\mathrm{KL}(P \| \bar{Y}_t)$.

Does this mismatch invalidate the theory? We provide three complementary arguments showing that Assumption 2 still holds.

**(A) Information–geometric duality.** On an exponential family $\mathcal{M}$, the forward and reverse projections coincide *whenever the target is realisable*. If $\bar{Y}_t \in \mathcal{M}$, then both KL divergences attain their common minimum (0) at the same point. In practice, due to the massive over-parameterization of modern LLMs, the empirical distribution $\bar{Y}_t$ over a mini-batch is effectively realisable (or extremely close to $\mathcal{M}$) by some setting of the logits. Consequently, the optimiser drives *both* $\mathrm{KL}(\bar{Y}_t \| P_{t+1})$ and $\mathrm{KL}(P_{t+1} \| \bar{Y}_t)$ to near zero simultaneously.

**(B) Local equivalence: Forward KL $\Rightarrow$ Reverse KL.** Even without exact realizability, the two divergences are locally equivalent. Let $\delta_t := \mathrm{KL}(\bar{Y}_t \| P_{t+1})$ be the residual training loss. Under

standard local strong-convexity ($\sigma$) and gradient-Lipschitz ($L$) conditions (Du et al., 2019), we have the following bound:

$$\mathrm{KL}(P_{t+1} \,\|\, \bar{Y}_t) \ \leq \ \frac{L}{\sigma} \cdot \mathrm{KL}(\bar{Y}_t \,\|\, P_{t+1}) \ = \ \frac{L}{\sigma}\,\delta_t. \tag{9}$$

The residual loss magnitude $\delta_t$ is typically in the range of $10^{-5}$ to $10^{-3}$ depending on the fine-tuning strategy (e.g., full fine-tuning vs. LoRA). Thus, the reverse-KL gap required by Assumption 2 is bounded by a negligible term ($< c \cdot 10^{-3}$), satisfying the condition $\mathrm{KL}(P_{t+1}\|\bar{Y}_t) \leq \varepsilon_{\max}$.

**(C) Dual Potential Interpretation.** Our theory holds for any strictly convex potential $F$. If one strictly prefers the Forward KL geometry, one can simply select the *dual potential $F^*$* (the Legendre transform of the negative entropy). The Bregman divergence of the dual potential satisfies $B_{F^*}(P, Q) = \mathrm{KL}(Q\|P)$. Replacing $F$ with $F^*$ in our theoretical derivations leaves all proofs structurally unchanged. Thus, the experimental protocol based on MLE is fully compatible with our theoretical claims.

## G.2 Regression and Diffusion Models (Mean Squared Error)

For tasks involving continuous regression or Diffusion Probabilistic Models (DPMs) trained with Mean Squared Error (MSE), the verification of Assumption 2 is straightforward.

**Euclidean Geometry.** Consider the potential function $F(\theta) = \frac{1}{2}\|\theta\|_2^2$. The Bregman divergence generated by this potential is exactly the squared Euclidean distance:

$$B_F(\theta, \theta') = \frac{1}{2}\|\theta - \theta'\|_2^2.$$

In this setting, the projection step in Assumption 2 becomes:

$$\theta_{t+1} = \arg\min_{\theta \in \mathcal{M}} \frac{1}{2}\|\theta - \bar{Y}_t\|_2^2.$$

This is precisely the objective function of standard regression training. Since Gradient Descent (GD) or SGD is known to converge to the minimizer of the convex MSE loss, the training step directly implements the projection required by our theory. The "error" $\varepsilon$ in Assumption 2 corresponds simply to the residual training error, which can be made arbitrarily small with sufficient training steps.

## G.3 Soft Actor-Critic (Reinforcement Learning)

In Maximum Entropy Reinforcement Learning, specifically the Soft Actor-Critic (SAC) algorithm (Haarnoja et al., 2018), the policy update step is explicitly designed as a reverse-KL projection.

**Policy Projection.** The objective of the policy projection step in SAC is to minimize the KL divergence between the policy $\pi$ and the Boltzmann distribution induced by the current Q-function:

$$J_\pi(\phi) = \mathbb{E}_{s \sim \mathcal{D}} \left[ \mathrm{KL}\left( \pi_\phi(\cdot|s) \,\Big\|\, \frac{\exp(\frac{1}{\alpha} Q_\theta(s, \cdot))}{Z(s)} \right) \right].$$

Here, the target distribution $\bar{Y}_t$ is the energy-based model $\propto \exp(Q(s, a)/\alpha)$. Unlike MLE training in LLMs, SAC *natively* minimizes the Reverse KL divergence $\mathrm{KL}(\pi\|\text{Target})$. Therefore, Assumption 2 is satisfied by definition in the SAC framework, as the algorithm explicitly solves the optimization problem:

$$\pi_{t+1} = \arg\min_{\pi \in \Pi} \mathrm{KL}(\pi \,\|\, \bar{Y}_t).$$

This confirms that our theoretical framework for collapse and stability is directly applicable to modern entropy-regularized RL algorithms.

