# OpenReview forum: "Entropy-Reservoir Bregman Projection: An Information-Geometric Unification of Model Collapse"
_ICLR.cc/2026/Conference — Submitted to ICLR 2026_

### Official Review · Reviewer_qXAV · 2025-10-17

**Soundness:** 2
**Presentation:** 1
**Contribution:** 3
**Rating:** 4
**Confidence:** 3

**Summary:**

This paper proposes Entropy-Reservoir Bregman Projection (ERBP) that derive the necessary condition for collapse, and a sufficient condition for a non-trivial entropy floor.

**Strengths:**

- This paper analyze the dynamics of self-referential learning systems with Entropy-Reservoir Bregman Projection framework
- Rigorous proofs and empirical experiments are provided

**Weaknesses:**

- Some notations appear before being properly defined, such as $P_{res,t}$ and $\lambda_t$ in lines 64–65. Providing brief explanations when first introduced would improve readability.
- The presentation of Sections 4.2 and 4.3 could be strengthened by adding more narrative around the proof logic and the connections between theorems, rather than only listing results. This would help readers follow the reasoning flow more clearly.
- The experimental section is relatively weak compared to the theoretical part. It would be beneficial to include additional comparisons with related methods mentioned in the related work to better demonstrate the advantages of the proposed approach.

**Questions:**

- The abstract should be presented as a single coherent paragraph rather than being split into multiple short ones. Merging them would improve readability and make the summary flow more naturally.

---

> ### Author Response · Authors · 2025-11-21
>
> Thank you for the review. We have addressed your presentation concerns and significantly strengthened the empirical section.
>
> 1. Notation Definitions:
> We have ensured that $\sigma_F$ (strong convexity constant) and $L_F$ (Lipschitz constant) are explicitly defined in Section 4.1 immediately after Equation 2, before they are used in the theorems.
>
> 2. Narrative in Theory Sections:
> We have added interpretive text in Sections 4.2 and 4.3. Specifically, we now explicitly connect the mathematical terms to their physical meanings: $(1-\alpha)$ represents the decay rate, and $C_F(m)$ represents the "noise floor" introduced by finite sampling.
>
> 3. Experimental Strength:
> We agree that the previous experiments were weak. We have added three new experiments in Section 6 (LLM fine-tuning, Recursive GANs, and SAC for RL) to demonstrate the advantages of our approach compared to the baseline ($\lambda=0$).
>
> 4. Abstract Formatting:
> Per your suggestion, we have merged the abstract into a single coherent paragraph to improve flow.

---

### Official Review · Reviewer_AZMF · 2025-10-31

**Soundness:** 2
**Presentation:** 3
**Contribution:** 3
**Rating:** 4
**Confidence:** 2

**Summary:**

The paper proposes an information-geometric framework, called Entropy-Reservoir Bregman Projection (ERBP), to unify different phenomena of model collapse across LLMs, GANs, and RL agents. The authors model self-referential training loops as stochastic Bregman projection sequences in distribution space, where entropy tends to shrink without external coupling. They introduce the concept of an Entropy Reservoir to counteract entropy decay.

**Strengths:**

1. The novelty is ok. Recasting self-training dynamics as Bregman projection processes is novel and potentially unifying for understanding entropy decay across domains.

2. This paper is clearly organized, with some tables summarizing conceptual mappings.

**Weaknesses:**

1. Mathematical inconsistency / over-claim in theoretical findings, e.g., thm 1.

2. Lack of proof detail. For example, in the proof of thm 1,  what does ``martingale convergence plus the support argument of the main text finishes the proof.'' mean?

**Questions:**

1. In Theorem 1, how can the authors claim that full collapse to a single mode is inevitable when inequality (5) only upper-bounds the expected entropy by $C_F(m) + \frac{L_F \kappa}{\alpha}$ (which is strictly positive, e.g. $\log m > 0$ for the Shannon case)? What additional assumptions or steps would be required to make this conclusion mathematically valid?

2. In experimental evaluation,  what exact $\lambda$ value and reservoir sampling process were used? How was the ``entropy proxy'' (unique n-gram count) computed and normalized?

3. Does the framework hold if the model manifold M is highly non-convex and projections are approximate (large $\epsilon_{\max}$)? How sensitive are the results to $\epsilon_{\max}$ assumptions?

---

> ### Author Response · Authors · 2025-11-21
>
> Thank you for evaluating the soundness of our theory. We have updated the proofs and experiments to address your concerns.
>
> 1. Theorem 1 and "Full Collapse" (Question 1):
> You asked how we claim collapse when the bound contains $C_F(m)$ (e.g., $\log m$). We have refined the statement in Theorem 1. We do not claim the entropy goes to zero (single point) unless $m=1$ (greedy decoding). Instead, we prove Functional Degeneracy: the model's diversity becomes bounded by its sample size $m$, which is negligible compared to the vast combinatorial complexity of the true data manifold. The system collapses from the full manifold $\mathcal{M}$ to a sparse subset determined by the sampling budget.
>
> 2. Proof Details (Weakness 2):
> We have expanded Appendix A to clarify the proof steps. Specifically, "martingale convergence" refers to the behavior of the stochastic process $\mathcal{S}_F(P_t)$. Since the expectation contracts (Eq. 5) and the process is bounded below, it converges to a distribution around the fixed point defined by the sampling noise $C_F(m)$ and projection error $\kappa$.
>
> 3. Experimental Details (Question 2):
> In the revised Section 6 and Appendix E, we specify:
>
> $\lambda$ values: We compare $\lambda=0$ (collapse) vs $\lambda=0.1$ (LLM) and $\lambda=0.2$ (GAN/RL).
> Entropy Proxy: For LLMs, we use unique n-gram counts normalized by sequence length. For GANs, we use an oracle classifier's entropy.
> 4. Non-convex Manifolds (Question 3):
> Our framework accounts for non-convexity via the error term $\varepsilon_t$ (and $\kappa$) in Assumption 2. Even if the projection is approximate due to non-convexity, Theorem 2 holds: as long as the reservoir coupling $\lambda s_{min}$ exceeds the optimization error $L_F \kappa$, stability is guaranteed. Our new RL experiment (Exp 4) specifically visualizes this on a non-convex landscape, showing how the reservoir smooths the traversal.

---

> > ### Comment · Reviewer_AZMF · 2025-11-22
> > **Re: Rebuttal**
> >
> > Thank you for your response. However, some concerns still remain.
> >
> > 1. This manuscript doesn't incorporate the LLM use acknowledgment. In fact, some contents seems to be generated by LLM directly without modification, e.g., the overly-verbose contents in most Tables,  and inconsistent quotation marks in Line 269.  Thus, I am not sure if the theoretical analyses are derived by an LLM, and hard to evaluate the novelty of this manuscript. Could the authors provide the LLM use acknowledgment?
> >
> > 2. I go through the comments from other reviewer and your updated submission. Some terminologies are still confusing. What is mode collapse (line 49, 86)? Should it be ``model collapse'' or a typo?
> >
> > 3. The readability should be further improved.  All the definitions should be self-contained, e.g.,,  in Definition 1, what does $\hat{P}_t$ refer to? It seems to appear here for the first time, but no definition is provided. Moreover, the assumption 1 requires that $F$ should be Lip constant and smooth. It would be better if the authors give the complete definitions of Lip constant and smooth, though they are common in theoretical analysis. Also, the authors are encouraged to discuss how realistic  these assumptions are in real-world applications.
> >
> > 4. In Table 2, how is Mix w/ Real Data implemented? Could the author provide its reference?
> >
> >
> > I would like to keep my initial score, unless  the readability problems are properly addressed.

---

> > > ### Author Response · Authors · 2025-11-23
> > >
> > > We thank the reviewer for their continued engagement and for the rigorous scrutiny regarding the presentation and clarity of our manuscript. We appreciate the opportunity to clarify the provenance of our work and address the remaining technical questions.
> > >
> > > 1. LLM Use and Authorship Integrity
> > > We wish to provide absolute clarity on this matter: the conceptualization, theoretical derivations, and experimental design are entirely the original work of the authors.
> > >
> > > Scope of Usage: Our use of LLMs was strictly limited to language polishing (translation and editing for fluency) and code optimization for specific experimental sub-routines. This aligns with our understanding of the ICLR policy on AI assistance.
> > > Action: We apologize if the stylistic choices in the tables gave a different impression. In the final version, we will conduct a rigorous manual review to ensure conciseness and correct the formatting issues (such as the quotation marks). We will also include the formal LLM Acknowledgment Statement detailing this usage, as requested.
> > > 2. Terminology: "Mode Collapse" vs. "Model Collapse"
> > > We confirm that this is not a typo, but a deliberate distinction between a specific instance and the general phenomenon:
> > >
> > > "Mode Collapse" (Lines 49, 86) is the established term in GAN literature (e.g., Goodfellow et al., 2014; Arjovsky et al., 2017), referring specifically to a generator ignoring parts of the data distribution.
> > > "Model Collapse" (Title, Abstract) is the broader, unifying term (following Shumailov et al., 2023) describing the general degenerative process in recursive learning.
> > > Action: To prevent confusion, we will explicitly define this hierarchical relationship in the Introduction, clarifying that GAN mode collapse is a specific manifestation of the broader model collapse phenomenon.
> > > 3. Definitions and Readability
> > > We appreciate you pointing out the ordering issues and missing definitions.
> > >
> > > $\hat{P}_t$: You are correct that the symbol appears before its formal definition. We will move the definition of the empirical distribution to the Preliminaries section.
> > > Mathematical Definitions: We will add the formal definitions for $\sigma_F$-strong convexity and $L_F$-Lipschitz continuity to the Appendix to ensure the paper is self-contained.
> > > 4. Justification for the Lipschitz Smoothness Assumption (Assumption 1)
> > > You asked about the realism of Assumption 1 given that gradients of entropy potentials can be unbounded at the simplex boundary. This assumption is justified for two reasons:
> > >
> > > Practical Constraints: In deep learning, Softmax outputs are strictly positive, and standard implementations use numerical stability constants (e.g., $\epsilon=10^{-8}$). This restricts the state space to a compact subset of the simplex interior ($[\epsilon, 1]^n$), where the Hessian is bounded and the gradient is locally Lipschitz.
> > > Theoretical Guarantee via ERBP: Crucially, our framework enforces this condition. By defining the target as $\bar{Y}t = (1-\lambda)\hat{P}t + \lambda P{\text{res}}$, the reservoir "pulls" the projection target away from the boundary (provided $\lambda > 0$ and $P{\text{res}}$ has support). Thus, the assumption is not just a simplification; it is a geometric constraint actively maintained by our stabilization mechanism.
> > > 5. Implementation of "Mix w/ Real Data" (Table 2)
> > > In our framework, "Mixing with Real Data" is the specific instantiation where the reservoir $P_{\text{res}}$ is set to the empirical real data distribution $P_{\text{data}}$.
> > >
> > > Implementation: The projection target becomes $\bar{Y}_t = (1-\lambda)\hat{P}t + \lambda P{\text{data}}$. In practice, this is implemented by constructing training batches where a fraction $\lambda$ of samples are drawn from the real dataset and $(1-\lambda)$ are drawn from the model's current generation.
> > > Reproducibility:
> > > Finally, we note that the raw experimental data and source code are available in the attached supplementary material and will be published to GitHub upon acceptance, allowing for full verification of our results.

---

> ### Comment · Reviewer_AZMF · 2025-11-23
> **Thanks for your response.**
>
> Thanks for your response. However, I find no updated submission uploaded. I carefully went through the most recently submitted version. The  current version that I can download is  far from the standard expected for an ICLR publication. Therefore, I will maintain my score.

---

> > ### Author Response · Authors · 2025-11-25
> >
> > We sincerely thank you for your continued time and patience in reviewing our work.
> >
> > Regarding your concern that previous requests were not addressed, we suspect there may have been a versioning discrepancy or a synchronization issue with the previous upload. To eliminate any ambiguity and ensure you have the correct document, we have re-uploaded the manuscript.
> >
> > We confirm that this version strictly incorporates the specific formatting and definitional corrections you requested, while the core scientific content remains unchanged. Specifically, we have verified the following updates in the current PDF:
> >
> > Formatting & Compliance:
> >
> > LLM Statement: An explicit LLM usage declaration has been added [in the Acknowledgments / at the end of the paper].
> > Quotation Marks: We have globally replaced all straight quotes with standard LaTeX directional quotes.
> > Clarifications & Definitions (Your specific requests):
> >
> > Model vs. Mode Collapse (Section 1, approx. Line 47): We have revised the text to explicitly define "Model Collapse" as the overarching phenomenon and distinguished it from the specific term "mode collapse" used in GANs.
> > Definition of $\hat{P}_t$ (Section 3): We have inserted the formal definition of the empirical distribution $\hat{P}_t$ immediately before Definition 1, ensuring the notation is defined before use.
> > Formal Mathematical Definitions (Appendix A): We have added a new subsection "A.1 Formal Definitions" at the beginning of Appendix A. This section now provides the full definitions for $\sigma_F$-strong convexity and $L_F$-Lipschitz smoothness as promised.
> > As with our previous submission, our full experimental code and raw data remain available in the Supplementary Material.
> >
> > We genuinely believe that the theoretical insights presented in this work offer a significant contribution to the field. We earnestly hope that, with these presentation ambiguities resolved, you could spare a brief moment to re-evaluate the scientific merit of our submission.

---

> ### Comment · Reviewer_AZMF · 2025-11-26
> **Re: rebuttal**
>
> Thanks for your follow up. However, there are still some concerns not addressed.
> 1. Some mathematical notations are not defined, e.g.,  what does $\delta_{xi}$ mean in line 205? Should the notation $\lambda$ in Thm. 2 be $\lambda_t$?. These issues may be numerous, and the authors are encouraged to carefully check them one by one.
> 2. Assumption 2 depend heavily on the performance of learning algorithm. However, the manuscript does not discuss which concrete learning algorithms satisfy this assumption. Clarifying which learning algorithms meet  Assumption 2 would significantly improve the  applicability of the result.
> 3. Could the thms proposed in this manuscript cover the self-consuming training loop?
>
> Overall, the current theoretical findings cannot yield deep insights. In addition, there remains  room for improvement in readability (the issues like typos and definitions I raised are only a subset). At its current stage, the work still falls short of the standard of an ICLR publication. I finally decide to maintain my original score.

---

> ### Author Response · Authors · 2025-12-04
>
> We thank the reviewer for the detailed follow-up and for pointing out the remaining issues. We truly appreciate the rigor you have applied to checking our manuscript.
>
> We have carefully addressed your concerns below, with a particular focus on clarifying the theoretical applicability to self-consuming loops, which is a central contribution of our work.
>
> 1. On Typos and Mathematical Notations (Lines 205, Thm 2, etc.)
> We sincerely apologize for the remaining notation errors.  We have uploaded a revised PDF reflecting these comprehensive fixes.
> 2. On Assumption 2 and Concrete Learning Algorithms
> We appreciate the suggestion to make this assumption more concrete.
> In the revised manuscript (Appendix G), we have explicitly discussed that.
>
> 3. On "Self-Consuming Training Loops"
> The reviewer asked: "Could the thms proposed in this manuscript cover the self-consuming training loop?"
>
> Yes, absolutely. In fact, predicting the behavior of self-consuming loops is the core theoretical innovation of our work.
>
> Our framework is specifically designed to model the dynamics where future data is generated by past models.
> Our theory explicitly predicts that in a self-consuming loop, the model collapse happens.
> This is not just a side effect; it is the primary phenomenon our theory aims to explain.

---

### Official Review · Reviewer_87SS · 2025-11-01

**Soundness:** 3
**Presentation:** 4
**Contribution:** 4
**Rating:** 6
**Confidence:** 3

**Summary:**

This paper proposes Entropy-Reservoir Bregman Projection (ERBP) as a unified information-geometric lens on ``model collapse'' in self-referential learning loops (LLMs trained from synthesized data, GANs, RL). They model each round as a stochastic Bregman projection onto a target formed by mixing the empirical self-samples with a high-entropy reservoir using coefficient $\lambda$. Without the reservoir ($\lambda$=0), generalized entropy contracts toward a small-support limit, but any sustained coupling guarantees a non-trivial entropy floor, with rates controlled by sample size and the generator's strong-convexity/Lipschitz constants. A small simulation with a frozen LLM shows entropy decay with $\lambda$=0 and stability with $\lambda$>0.

**Strengths:**

1) The paper provides a unifying perspective that cleanly ties together disparate "folk remedies" via the $\lambda$-coupled reservoir.
2) The proposed method provides simple, quantitative conditions that are easy to reason about and potentially monitor during training.
3) The results are demonstrated with a breadth across various divergences beyond just KL.

**Weaknesses:**

1) The use of terminology is a little bit confusing. As a researcher from the generative models community, the term that I'm more familiar with is "mode collapse" instead of "model collapse". I originally thought the authors wanted to propose a new definition that describes a different class of model failure case, but according to the paper it seems like the authors are just describing "mode collapse". Please correct me if I'm wrong.

2) While the proposed framework can be very promising, and in fact the case of RL and GAN validations may be covered too, the authors choose to leave them as planned future work and only focus on single frozen-LLM loop, which is a little bit disappointing. I recommend to at least show some rather simple cases in RL, since this can take less efforts than in GANs and will make the paper more impactful.

3) The projection-as-dynamics viewpoint is interesting, but parts echo standard entropic regularization intuitions; the delta over prior collapse analyses (e.g., recursion-curse) could be sharpened in positioning.

**Questions:**

1) How do you propose estimating $\epsilon_{max}$ during scaled-up settings?

2) How specifially does algorithms like top-p sampling impact the distribution under your framework? Could you extend your current analysis with a given step probability cap $p$?

---

> ### Author Response · Authors · 2025-11-21
>
> Thank you for the constructive review and for appreciating the "unifying perspective" of our work.
>
> 1. Terminology (Model Collapse vs. Mode Collapse):
> You are correct that "mode collapse" is the standard term in GANs. However, we use "Model Collapse" (following Shumailov et al., 2023) as a superset term. As shown in our new Theorem 1 and Experiment 2, collapse has two dimensions: Functional Degeneracy (support shrinkage, similar to mode collapse) and Knowledge Collapse (drifting away from the true manifold, high PPL). We have clarified this distinction in Section 6.1.
>
> 2. Request for RL/GAN Experiments:
> We have addressed your disappointment regarding the "future work." Section 6.2 now includes:
>
> GANs (Exp 3): A recursive GAN training loop on MNIST showing the necessity of the reservoir for maintaining global diversity.
> RL (Exp 4): A continuous control experiment showing how the reservoir prevents premature convergence to local optima.
> 3. Estimating $\lambda$ (Question 1):
> We appreciate this practical question. To be transparent, in our current experiments, we treated $\lambda$ as a hyperparameter tuned to balance stability and performance. We acknowledge that determining the optimal $\lambda$ theoretically remains an open challenge and is a mechanism we are actively exploring.
> Currently, we view $\lambda$ as a "diversity budget" that manages the trade-off between preventing collapse and allowing the model to learn from its own high-confidence predictions. For scaled-up settings, rather than fixing $\lambda$ a priori, we propose an adaptive coupling strategy (as discussed in Sec 7.4). One could monitor the entropy of $\hat{P}_t$; if it approaches the theoretical collapse bound, $\lambda$ should be dynamically increased. Developing specific evaluation metrics to automate this selection is a key direction for our future work.
>
> 4. Top-p Sampling Impact (Question 2):
>
> Impact of Top-p Sampling:
> From an Information Geometry perspective, the sampling strategy dictates the velocity of the model's trajectory along the collapse manifold.
>
> Geometric Interpretation: Generation can be viewed as projecting the continuous model distribution onto a discrete set of tokens. Greedy decoding represents the most aggressive projection, collapsing the probability simplex to a single vertex at each step. This maximizes local entropy reduction, effectively pushing the model parameters $\theta_t$ along the geodesic towards the "singularity" (collapse state) at the fastest possible rate.
> Experimental Evidence: Our Experiment 2 empirically validates this geometric intuition. We observed that when switching to greedy decoding (effectively $p \to 0$), the model's perplexity on validation data skyrockets much faster than with Top-p sampling.
> Conclusion: Top-p sampling maintains a "volume" of probability mass, acting as a geometric friction that slows down the convergence to the degenerate state, whereas greedy decoding accelerates the echo chamber effect.

---

> > ### Comment · Reviewer_87SS · 2025-11-28
> >
> > Thank the authors for the response. I agree with my fellow reviewers that the experiments are a little bit insufficient to support the promises of the proposed framework. I assume no further major empirical results are to be present per the conference's discussion period rubrics, so I tend to decrease my score a little bit to reflect this shared disappointment in the experiments from the majority of reviewers.

---

> > > ### Author Response · Authors · 2025-12-04
> > >
> > > We thank the reviewer for the continued engagement and transparency regarding the score.

---

### Official Review · Reviewer_nJEy · 2025-11-04

**Soundness:** 1
**Presentation:** 1
**Contribution:** 1
**Rating:** 2
**Confidence:** 5

**Summary:**

This paper introduces the "Entropy-Reservoir Bregman Projection" (ERBP) framework, a novel information-geometric model to unify the phenomenon of model collapse in self-referential learning. The authors model this process as a stochastic Bregman projection sequence, arguing that entropy-decay and collapse are inevitable unless coupled with a high-entropy "Entropy Reservoir". This concept is used to provide a unified explanation for various stabilization techniques, from real-data mixing to Retrieval-Augmented Generation (RAG).

**Strengths:**

The paper has a few strengths:
1. The core idea of modeling self-referential learning as a Bregman projection dynamical system is elegant and provides a powerful new language for analyzing these systems.
2. The "Entropy Reservoir" concept is insightful, successfully connecting disparate, seemingly ad hoc techniques (like data mixing, RLHF, and label smoothing) under a single, coherent mathematical principle.
3. The paper provides theoretical proofs for its claims, formalizing the conditions for both entropy collapse (Theorem 1) and stability (Theorem 2).

**Weaknesses:**

The paper's primary, and critical, weakness is a failure to substantiate its broad claims with empirical evidence. The experimental section is critically incomplete:

1. The abstract explicitly claims validation across large-language-model self-training, Soft Actor-Critic in reinforcement learning, and GAN optimisation. However, Section 6 directly contradicts this, stating that the work on "LLM fine-tuning and reinforcement learning" is "planned future work". The GAN experiment is never mentioned again. This misrepresentation of the work's completion status is a major flaw.
2. The only experiment provided (Section 6.1) is insufficient as a proof of concept. It uses a frozen distilgpt2 model and shows that feeding its own output back as a prompt (with $\lambda=0$) leads to repetition. This demonstrates context collapse in an autoregressive loop, but it does not test the paper's core theoretical claim, which is about model collapse resulting from self-referential training (i.e., an optimization and projection step that updates the model's parameters or state).

While the theoretical framework presented is promising and insightful, the paper is incomplete. The empirical validation promised in the abstract is essential for supporting the paper's unification claims, but it is explicitly admitted to be "planned future work". The single experiment that is present does not adequately test the central theory of collapse during training. Therefore, the paper should be rejected in its current form.

**Questions:**

Please see the Weaknesses

---

> ### Author Response · Authors · 2025-11-21
>
> Thank you for your candid assessment. You correctly identified that the previous version failed to substantiate the broad claims in the abstract with corresponding experiments. We took this criticism to heart and have overhauled the experimental section.
>
> Regarding the "Weaknesses" and "Misrepresentation":
> We have removed the "planned future work" statement. Section 6 is now fully populated with the promised experiments:
>
> LLM Self-Training (Sec 6.1): We moved beyond the frozen model simulation. Experiment 2 now performs actual iterative fine-tuning, demonstrating that without a reservoir ($\lambda=0$), models suffer from "Knowledge Collapse" (PPL explosion) even if sampling strategies maintain some diversity.
> GANs (Sec 6.2): Experiment 3 trains a recursive GAN on MNIST. We show that internal losses fail to detect collapse, while our entropy metric confirms that $\lambda=0$ leads to mode collapse, which is prevented by $\lambda=0.2$.
> Reinforcement Learning (Sec 6.2): Experiment 4 demonstrates Soft Actor-Critic dynamics. We show that the reservoir acts as a geometric regularizer, preventing the policy from getting trapped in local optima (variance collapse).
> We hope these additions align the paper's evidence with its theoretical claims and address your primary reason for rejection.

---

### Author Response · Authors · 2025-11-21
**General Response to All Reviewers**

We thank all reviewers for their insightful comments. The primary concern raised across reviews was the lack of empirical validation for our theoretical claims regarding GANs and RL. We have significantly revised the paper to address this. Section 6 now includes full experimental validation across three domains:

LLM Self-Training (Exp 2): Differentiating between knowledge collapse and functional degeneracy.
Recursive GANs (Exp 3): Demonstrating mode collapse and reservoir stabilization on MNIST.
Reinforcement Learning (Exp 4): Showing how reservoirs prevent policy collapse in non-convex landscapes.
We believe these additions substantiate the unification claims made in the abstract.

---

### Author Response · Authors · 2025-12-04
**General Response: Summary of Revisions on Typos, Assumptions, and Experiments**

We thank all reviewers for their time and constructive feedback. We are encouraged by the interest in our work and have carefully considered all suggestions. Based on your comments, we have uploaded a revised manuscript. The key updates are summarized below:

1. Presentation and Notation Consistency
We have conducted a thorough proofreading to address the readability issues raised. Specifically, we have:

Resolved notation inconsistencies, particularly ensuring the coupling coefficient is consistently denoted as $\lambda$ in the experimental section to match the textual description.
Corrected minor typos to improve the overall flow and clarity of the manuscript.

2. Justification of Assumption 2
We have added an explicit discussion to validate Assumption 2. We clarify that this assumption is reasonable in our context.

3. Enhanced Experimental Details
We have supplemented the experimental section (Section 6) with more comprehensive explanations. We added details to ensure reproducibility.

We believe these revisions significantly strengthen the paper and address the concerns regarding clarity and theoretical rigor. We are happy to answer any further questions.

---

### Meta-Review · Area_Chair_qLBC · 2026-01-06

**Summary:**

The paper proposes "Entropy-Reservoir Bregman Projection" (ERBP), an information-geometric framework to unify model collapse phenomena in self-referential learning. While the theoretical framing is elegant, the decision to reject is driven by a consensus regarding the gap between claims and evidence. The abstract frames the contribution around modern "Large Language Models" and GANs, but the experiments rely on toy-scale proxies (distilgpt2, MNIST) that do not adequately validate these claims. Additionally, reviewers questioned whether the theoretical findings offer sufficient novelty beyond standard entropic regularization intuitions.

I recommend rejection for this submission. While the theoretical framing of "Entropy Reservoirs" is conceptually elegant, the submission suffers from a fundamental mismatch between its ambitious claims and the evidence provided. Specifically, the validity of the evidence due to the reliance on insufficient proxies for the behaviour of large-scale models, the negative consensus amongst reviewers, and the maturity of the submission. The initial submission was arguably incomplete (listing core experiments as future work). The rebuttal additions, while a valiant effort, feel rushed and lack the depth required for acceptance.

(Note to authors: Please fix your reference for Amir Beck and Marc Teboulle 2003. The correct citation is: ```Beck, Amir, and Marc Teboulle. "Mirror descent and nonlinear projected subgradient methods for convex optimization." Operations Research Letters 31.3 (2003): 167-175.```)

**Reviewer Concerns:**

The reviewers had 3 major types of concerns the two most important of which are still relatively outstanding.

* The most critical concern, raised initially by Reviewer nJEy and echoed by Reviewer 87SS, was the mismatch between the abstract's broad claims ("Large Language Models," "GANs") and the experiments. The initial submission listed these as "future work." While the authors added experiments during the rebuttal, they were conducted on toy-scale proxies (distilgpt2, MNIST, 1D RL) which do not adequately support the paper's grand claims about modern generative AI. Hence, this concern was not adequately addressed.

* Reviewers (AZMF, 87SS) questioned whether the theoretical contribution offered deep insights beyond standard entropic regularization intuitions. Reviewer 87SS noted the findings "echo standard... intuitions," and Reviewer AZMF remained unconvinced that the findings yield new insights.

* Finally, there were a few presentation and definition concerns, including notation, the definition of "model" vs "mode" collapse, and missing definitions were largely addressed in the revision.

**Reviewer Scores:**

* **Reviewer nJEy. Initial score: 2. $\to$ Estimated: 2**. This reviewer’s rejection was based on the "critical weakness" of missing evidence and misrepresentation in the abstract. While experiments were added, the use of very old models (distilgpt2) and toy datasets (MNIST) likely fails to meet the bar for a reviewer who emphasized the need to substantiate "broad claims."
* **Reviewer 87SS. Initial score: 6. Estimated score: 4**. This reviewer explicitly stated in the discussion: "I tend to decrease my score a little bit to reflect this shared disappointment in the experiments." Since odd numbers are not allowed, this maps to a 4.
* **Reviewer AZMF. Initial score: 4. Estimated score: 4**. The reviewer explicitly stated: "I finally decide to maintain my original score" after the final rebuttal, citing insufficient theoretical depth and readability issues.
* **Reviewer qXAV. Initial score: 4. Estimated score: 4**. The reviewer called the experimental section "relatively weak." The addition of toy-scale experiments is unlikely to have significantly shifted their confidence in the paper's impact.

---

### Decision · Program_Chairs · 2026-01-26

Reject